# Tissue Culture Innovations for Propagation and Conservation of Myrteae—A Globally Important Myrtaceae Tribe

**DOI:** 10.3390/plants13162244

**Published:** 2024-08-13

**Authors:** Jingyin Bao, Billy O’Donohue, Karen D. Sommerville, Neena Mitter, Chris O’Brien, Alice Hayward

**Affiliations:** 1Centre for Horticultural Science, Queensland Alliance for Agriculture and Food Innovation, The University of Queensland, St. Lucia, QLD 4072, Australia; billyodonohue@gmail.com (B.O.); n.mitter@uq.edu.au (N.M.); c.obrien4@uq.edu.au (C.O.); 2Australian Institute of Botanical Science, The Royal Botanic Gardens and Domain Trust, Mount Annan, NSW 2567, Australia; karen.sommerville@botanicgardens.nsw.gov.au

**Keywords:** ex situ conservation, tissue culture, field gene banks, cryopreservation, Myrteae

## Abstract

Myrteae is the most species-rich tribe in the Myrtaceae family, represented by a range of socioeconomically and ecologically significant species. Many of these species, including commercially relevant ones, have become increasingly threatened in the wild, and now require conservation actions. Tissue culture presents an appropriate in vitro tool to facilitate medium-term and long-term wild germplasm conservation, as well as for commercial propagation to maintain desirable traits of commercial cultivars. So far, tissue culture has not been extensively achieved for Myrteae. Here, tissue culture for *Eugenia*, one of the most species-rich genera in Myrteae, is reviewed, giving directions for other related Myrteae. This review also focuses on ex situ conservation of Australian Myrteae, including using seed banking and field banking. Despite some progress, challenges to conserve these species remain, mostly due to the increasing threats in the wild and limited research. Research into in vitro methods (tissue culture and cryopreservation) is paramount given that at least some of the species are ‘non-orthodox’. There is an urgent need to develop long-term in vitro conservation for capturing the remaining germplasm of threatened Myrteae.

## 1. Introduction

The family Myrtaceae, as the ninth-largest angiosperm family, is represented by about 6000 species, 146 genera, and 17 tribes [1,2,3]. The Myrtaceae species are widely distributed from tropical to warm temperate regions, centered in South America and Australia [4]. Myrteae is the most species-rich monophyletic tribe in the family and comprises 51 genera and about 2500 species, accounting for more than a third of all the Myrtaceae species [2,3,5]. The species in this tribe are very diverse, but most belong to two large genera—*Eugenia* (1219 spp.) and *Myrcia* (782 spp.) [6,7]. 

Myrteae is reported to have great ecological significance in the neotropics, where the species are mostly distributed. *Eugenia*, for example, is recorded as the most abundant genus in Atlantic forests [8]. In these forests, pollinators are attracted to the aggregated flower patterns of *Eugenia* and other Myrteae/Myrtaceae spp. and successful pollination provides fruits of varying sizes throughout the year for frugivorous birds, bats, and monkeys [9,10]. In Brazil, a species called *Eugenia uniflora*, more commonly known as Brazilian cherry, can survive on degraded or disturbed sites and is essential for maintaining local vegetation affected by human activity [11]. This species and a number of others in Myrteae also have economic value and are used to produce edible fruits, essential oils, and medicine [12,13]. Familiar fruits include guava (*Psidium guajava*), feijoa (*Acca sellowiana*), jaboticaba (*Plinia cauliflora*), and Brazilian cherry [14,15]. *Myrtus communis*, a cultivated ornamental plant with multiple economic values, is a species valued by food and pharmaceutical industries for its culinary and medicinal properties [16]. In some cultures, it is also used as a ritual plant at births, weddings, and funerals to symbolize eternity, purity, and regrowth [17]. These examples highlight not only the economic value of the Myrteae, but also the cultural and spiritual values held for some of these species.

Despite their value, Myrteae species have been increasingly threatened by anthropogenic activities that lead to deforestation and habitat fragmentation, as well as natural disturbance [18,19]. For instance, it is estimated that only 11.4–16% of the original Brazilian Atlantic Forest remains, less than 20% of which remains unfragmented and is home to many Myrteae species [20,21,22]. The richness of Myrteae in the Brazilian Atlantic Forest has been highly reduced due to the decreasing forest cover, with potential to interrupt associated ecological systems [23,24]. Furthermore, many studies have confirmed that neotropical forest ecosystems are vulnerable to extreme weather, resulting in negative impacts on tree species composition and species populations [25,26,27,28]. 

In addition, Myrteae species are also susceptible to the fungal pathogen *Austropuccinia psidii*, causal agent of the disease known as myrtle rust [29]. *A. psidii* originated in neotropical forests and has spread globally, including to Australia and New Zealand, where it is having major negative impact [30,31]. Myrtle rust primarily impacts the young tissues of the plant, including fruit and flowers [32,33], and can lead to defoliation, shoot dieback, and eventual mortality, depending on the susceptibility of different species [34,35]. The pathogen has a very broad host range of around 73 genera in the family Myrtaceae, including *Eugenia* and *Myrcia* [36,37,38]. Myrteae species are particularly impacted because their high distribution in warm and humid environments, including Pacific islands and neotropical regions, favors the development of myrtle rust [34,39]. Since its first detection in Australia in 2010, four Myrteae taxa (*Rhodamnia rubescens*, *R. maideniana*, *Rhodomyrtus psidioides*, and *Lenwebbia* sp. Main Range) have been declared critically endangered as a direct result of the disease [34]. Many other Myrteae taxa are also on a trajectory towards extinction [29].

Due to these threats, many Myrteae species are recognized as rare or endangered. In South America, the International Union for Conservation of Nature (IUCN) red list recognizes more than 200 Myrteae species as endangered or critically endangered, most of which are from the genera *Myrcia* and *Eugenia* [40]. In Australia, there are 14 threatened Myrteae species in the Southeast Queensland region alone, including *Gossia*, *Lenwebbia*, *Rhodamnia*, *Rhodomyrtus,* and *Uromyrtus* species [41]. Ex situ conservation is proving to be essential to preserve these species [42,43]; however, many of them fall in the category of ‘exceptional’ and are not suitable for seed banking [44]. Moreover, seed production and germination are poor, for some of these species, due to disease or reproductive barriers. Long-term ex situ conservation therefore requires other methods, such as tissue culture micropropagation and cryopreservation. 

Little attention has been paid to systematically reviewing the tissue culture protocols of broader Myrteae, which includes Australia’s most imperiled myrtle rust-impacted genera and various species that are ecologically, culturally, and economically important. Exceptions are guava and feijoa, which have long been investigated and recently reviewed due to their economic value [45,46].

This review summarizes available tissue culture protocols for *Eugenia* species, one of the largest and most prominent genera reported in the literature. It provides an overview of protocols utilized for different purposes, from propagation to conservation. It also covers the current conservation efforts done for the Australian Myrteae. As tissue culture technology forms the basis for in vitro conservation methods, as well as efficient amplification of plants suffering poor natural reproduction, this review aims to provide a key resource for research into propagation and conservation of broader Myrteae.

## 2. Micropropagation of Myrteae

### 2.1. Introduction to Micropropagation

Plant tissue culture is a widely utilized tool across a range of industry and research applications. Most commonly, tissue culture is designed for the rapid generation of plant propagules, a practice known as micropropagation, which has significant potential in a range of applications, including horticulture (of ornamentals and fruit crops), forestry, and conservation science [47,48,49,50]. As is summarized by Cardoso, et al. [51], micropropagation is the preferred method for propagating taxa with (1) problems in sexual or vegetative reproduction; (2) a limited number of mother plants; (3) a disease that can be eliminated by in vitro culture; (4) a requirement for production of only one sex; or (5) a need for ex situ conservation. It can also be used to facilitate multiplication of elite clonal varieties and hybrid development. For instance, approximately 156 genera of ornamental plants have been reported to be propagated or genetically improved using tissue culture technologies where desirable traits, such as scent and color, can be manipulated to meet market demand [50,52,53]. The incorporation of genetic knowledge into tissue culture can improve the efficiency of breeding programs through the use of trait-related genetic markers for early and targeted screening [54,55], genetic transformation [56,57], and genome editing [58]. With specific reference to plant conservation, micropropagation offers the unique advantage of rapid, mass-propagation from a very limited supply of mother plant material, a challenge often associated with endangered or at-risk species. Several studies have demonstrated the utility of micropropagation for the conservation of endangered species [59,60,61] and difficult-to-propagate species [62]. For conservation purposes, tissue culture should also be combined with genetic diversity capture and screening to ensure that diversity is represented in the tissue cultured population [63]. Tissue culture is also a prerequisite for ex situ conservation methods such as cryopreservation, where regeneration of cryo-suspended plant material is required [64].

However, despite its relatively high efficiency, micropropagation protocols are not widely available for woody plants due to their common recalcitrance to tissue culture. These protocols are typically species- and even cultivar-specific and require extensive research to develop. Tissue culture also requires initial capital expenditure on infrastructure and a high input of skilled labor, electricity, and consumables compared to conventional propagation methods [51,61,65].

### 2.2. Tissue Culture for Micropropagation of Myrteae

Tissue culture development for Myrteae species has focused primarily on economically viable species, with some work on highly threatened species requiring ex situ conservation. Ornamental plants such as common myrtle, which were previously propagated by seed or cuttings, are now extensively propagated by tissue culture to maintain desired genotypes for indoor or garden growing [66]. Tissue culture has also been incorporated into the production of fruit trees in Myrteae, either by facilitating seed germination or optimizing production efficiency [67,68]. This is due to the variability of sexual reproductive capacity of Myrteae species among genera and species [69,70]. Reproductive success is determined by a series of processes including pollination, fruit (seed) development, germination, and establishment, and is limited by factors such as low pollination efficiency, predation of flowers, fruit or seed, and poor germination [71]. Seed setting is very rare for some threatened Myrteae species, such as *G. gonoclada*, due to small fragmented populations limiting pollination success, and failure of some fruit to ripen [18]. Given that myrtle rust can infect reproductive parts of many Myrteae plants, such as *Decaspermum humile* and *Rhodamnia rubescens*, this also reduces seed production and further impacts population growth [72]. Several *Eugenia* species, including *E. uniflora* and *E. brasiliensis*, show limited seedling survival after germination; this is thought to be a result of self-inhibitory compounds secreted by earlier germinated seedlings [73,74]. Likewise, the germination of *Rhodomyrtus tomentosa* seeds collected from different locations was found to be slow and erratic even under optimal temperatures [75]. Vegetative propagation can occur naturally (through suckers) or can be induced via cutting propagation methods to replicate Myrteae plants. Some Myrteae species are readily amenable to vegetative propagation [43]. However the success of vegetative propagation can vary greatly with the age and type of material used [76], the timing of collection, and the health of the plant [43]. Comparably, tissue culture enables rapid multiplication at any time of year, avoids the impacts of pests, disease, and other environmental stresses, and is valuable to long-term ex situ conservation of species not suitable for seed banking. In this literature review, advancements in tissue culture for *Eugenia* are showcased due to its diversity and species-richness to provide a guide for further Myrteae tissue culture development.

### 2.3. Tissue Culture of Eugenia Species

#### 2.3.1. Establishment

Establishing sterile cultures is the first step in developing an efficient micropropagation protocol, often referred to as tissue culture initiation. Theoretically it is feasible to introduce any plants into tissue culture; however, achieving sterile plant materials or ‘explants’, and subsequent vigor of the initiated explant, is highly dependent on the condition of the mother plant [65,77]. Mother plants used as the source of explant material are often cultivated under conditions promoting initiation success, such as in glasshouses where temperature and light conditions as well as pathogen exposure can be largely controlled [78]. They are often pretreated with disinfectants to improve health [79,80]. For example, de Assis, et al. [81] reported spraying the antibiotic penicillin and a systemic fungicide onto *E. pyriformis* mother plants before initiating nodes into cultures. Similarly, mother plants pretreated with the fungicides thiophanate-methyl and streptomycin sulphate were used to extract nodal segments for establishing *E. involucrata* cultures [82]. Hormone pretreatments have also been reported to improve morphogenesis and vigor in initiated *Eugenia* cultures. Spraying 6-benzylaminopurine (BAP) (1 mg L^−1^) solution onto mother plants twice a week for 4 weeks was reported to stimulate shoot growth in *E. anthacanthoides*; however, the shoots were then more sensitive to sodium hypochlorite (NaClO) than those obtained from the mother plant without pretreatment [83]. 

Explants are often sourced from regions of the plant-harboring meristematic tissue and that are actively growing, such as buds or shoot tips [84]. These meristematic cells are undifferentiated and more readily manipulated through hormone exposure. They are also more likely free from internal pathogens due to the lack of a developed vascular system [85]. Commonly used explants for *Eugenia* micropropagation include seeds, embryos, shoots, axillary or nodal buds, and meristems (Table 1). Seeds are the most commonly used starting material for micropropagating *Eugenia* species. The use of seeds to establish tissue cultures is beneficial for conservation purposes as each seed is genetically distinct [65]. This is in contrast to cuttings that will be clonal to mother plants. Seeds are sometimes germinated in vitro because of the germination difficulties encountered in natural environments, as was the case for *E. sulcata* [86]. Moreover, in vitro germinated seeds are capable of producing different forms of uncontaminated explants, and the juvenility of these explants makes them highly responsive to tissue culture, leading to high shoot proliferation and rooting capacity [87,88]. Seed maturity has been shown to influence in vitro germination potential and further seedling growth. A study on *E. uniflora*, for example, showed that seeds from unripened fruits generally had a lower dormancy rate and initially generated slightly shorter seedlings after germination than the mature seeds [89]. The possibility of germinating immature, cut, or even damaged domesticated or endangered *Eugenia* seeds has been proven, allowing wider use of seeds of different qualities as starting material for tissue culture [90,91]. 

Explants derived from vegetative material, such as shoots, meristems, and nodes, are also very commonly used in *Eugenia* tissue culture. They can be used to establish true-to-type cultures useful for propagating elite commercial *Eugenia* cultivars [92]. However, adventitious shoots generated from callus in shoot culture may be genetically different to the original explants because callus cells are not necessarily genetically homogeneous [65]. The maturity of the plants from which explants are collected has an impact on their micropropagation potential, with explants from older plants having lower regeneration capacity [88,93]. A method used to tackle this problem is to rejuvenate the plants before initiating in vitro cultures. This can be done using a variety of stress factors, such as grafting, or non-stress factors, such as changing pH and inducing DNA methylation [94]. 

For initiating *Eugenia* species, the most commonly used disinfectant agents include ethanol, bleach, mercuric chloride (HgCl_2_), and commercial detergents (Table 1). The goal of surface sterilization is to neutralize bacterial and fungal pathogens, without causing toxicity to the explants [95]. These disinfectants are usually used in series to improve the sterilizing efficiency, with sterilized water rinsing between each treatment. Bleach, either sodium hypochlorite (NaOCl) or calcium hypochlorite (Ca(OCl)_2_), is normally used in the concentration of 0.3–3% active chlorine for 10 to 25 min immersion (Table 1). Surfactants, including Tween 20, are usually added to the bleach in small amounts (0.01–10%) to improve wetting of the explants and the effectiveness of sterilization [83,96]. HgCl_2_ has been historically utilized as an explant disinfectant due to its highly effective antimicrobial action; however, it is becoming less common due to damage caused to plant tissue and the human health hazard of mercury exposure [97,98,99]. Gallon, et al. [100] have also reported the application of a novel surface sterilization agent—metal nanoparticles—to disinfect 1.5 cm *E. involucrata* shoot segments extracted from seedlings that germinated in soil-free substrate under controlled environmental conditions. This study showed that treatment with silver nanoparticles resulted in the lowest contamination (4.16%) and did not hamper further shoot proliferation, compared with sterilization using other metal nanoparticles, antibiotic treatment, or the most used ethanol–bleach treatment. However, despite proper surface sterilization, endophytes (microorganisms that are less sensitive and stay inside plant tissues) may still survive the process and be retained in the explants [101]. Although they have not been reported as a problem in *Eugenia* tissue culture, this causes problems, such as reduced culture viability and rooting difficulties, in many in vitro cultures [102,103].

Media used to establish *Eugenia* cultures vary based on the type of starting material. For in vitro seed germination, solid water agar, Murashige and Skoog (MS) medium [104], and Woody Plant Medium (WPM) [105], are the most used basal media, with pH adjusted to 5.7 to 5.8. These basal media are sometimes used in reduced strength to improve germination efficacy, as shown for such species as *Withania somnifera* [106], *Aloe polyphylla* [107], and *Pterocarpus marsupium* [108]. Carbohydrates and plant hormones may also be included in in vitro seed germination media to provide a carbon source and to regulate plant regeneration [109]. It is reported that cytokinins participate in all phases of seed germination of many species, including *Arabidopsis* and *Lotus* [110,111,112]. Studies have also shown that exogenous cytokinins are able to promote seed germination by alleviating abiotic stress, such as those caused by salinity [113]. Blando, et al. [114] reported the use of a MS basal medium containing half-strength macronutrients, full-strength micronutrients, and 2.5 μM thidiazuron (TDZ) as the cytokinin, to germinate *E. myrtifolia* seeds in the dark. In most cases, intact *Eugenia* seeds are used for germination in tissue culture, but sometimes the integument is removed to improve access to the nutrients in the medium and to improve regeneration capacity [114]. For seed culture, one type of medium is usually used throughout the initiation, multiplication, and rooting process. MS basal medium supplemented with 3.8 μM thiamine and 555.1 μM myo-inositol has been used for successful germination of *E. anthacanthoides* and *E. subdisticha* seeds [83]. For *E. uniflora*, Griffis [115] suggested using water agar for seed germination and supplementing the nutrients every 4 weeks with half-strength liquid WPM for shoot multiplication and rooting.

Similar media to those used for seeds have been used to initiate cultures with shoot nodes or shoot tips as starting materials. For example, Toussaint, et al. [116] reported using an MS basal medium supplemented with 2.2 μM BAP as the cytokinin and 0.5 μM indole-3-butyric acid (IBA) as the auxin to initiate *E. smithii* shoot tip cultures. However, some *Eugenia* tissue culture protocols used cytokinin alone without auxins during initiation to stimulate lateral growth and prepare for further multiplication. Uematsu, et al. [117] established *E. uniflora* cultures using MS medium with 0.9 μM BAP and incubation in the dark to encourage shoot tip regeneration. Antioxidants and other anti-browning agents are also commonly added into initiation media to reduce the release of phenolic compounds and prevent oxidation [118,119]. Longo, et al. [120] used polyvinylpyrrolidone (PVP) and ascorbic acid to initiate *E. myrtifolia* in vitro cultures using spring buds. PVP and ascorbic acid may also be incorporated into multiplication media to control oxidative browning [121]. Application of PVP and ascorbic acid, alone or in combination, have been reported extensively in pretreatment of explants and in media to improve tissue culture quality of woody species (for example, *Brahylaena huillensis* [122] and *Punica granatum* [123]) and non-woody species (for example, *Vicia faba* [124] and *Paeonia lactiflora* [125]). 

Meristem culture is a tissue culture technique that isolates and cultures the shoot apex of a donor plant and is beneficial for propagating virus-free plants with high multiplication efficiency [126]. Meristem culture of *Eugenia* species generally uses liquid media. Wang and Charles [127] pointed out that liquid media supported better results for meristem culture than solid media, and suggested this may be due to the dilution of phenolic metabolites. Kataoka and Inoue [128] reported using meristem culture to overcome propagation difficulties (e.g., seasonal fluctuation) and improve micropropagation efficiency of *E. javanica*. They used liquid MS medium with 2.2 μM BAP and 87.6 mM sucrose for the first 4 weeks of initiating meristem culture. 

#### 2.3.2. Multiplication

Cytokinins, together with low levels of auxins, are often used in combination to promote shoot multiplication of plants in tissue culture. Multiplication of *Eugenia* cultures, regardless of the explant type used for initiation, generally relies on full-strength or half-strength MS media with auxins and cytokinins normally added in an individually tailored ratio (Table 1). The most commonly used pair of plant growth regulators is BAP and IBA, commonly combined in a ratio of 1.8:1 [100,116,129]. However, an exception was the tissue culture protocol of *E. myrtifolia* developed by Longo, et al. [120], which reported using 2.2 μM BAP and 0.05 μM IBA in the multiplication media as well as 0.1 μM gibberellic acid (GA_3_). There was also one micropropagation protocol of *E. dysenterica* that included cytokinin (BAP), but without auxin, in WPM for multiplication [130]. Other cytokinins, such as thidiazuron (TDZ), and auxins, such as 1-naphthaleneacetic acid (NAA), have sometimes been used in place of the BAP and IBA combination [82,128]. Hormone-free media, or steps, may also be used during shoot regeneration. For instance, a half-strength MS basal medium without hormones was used to promote shoot elongation after *E. uniflora* shoot tip regeneration because vitrified shoots were observed on media containing BAP, despite higher proliferation rates [117].

#### 2.3.3. Rooting and Acclimatization

Rooting of *Eugenia* cultures can generally be classified into two types: spontaneous rooting and induced rooting. Several *Eugenia* species, including *E. anthacanthoides*, *E. subdisticha,* and *E. sulcata*, have been reported to develop roots on a single medium used throughout the process from initiation to rooting [83,86]. Likewise, spontaneous rooting has been observed for *E. uniflora* and *E. involucrata* in vitro shoots on multiplication media [100,117]. It was observed that *E. myrtifolia* cultures rooted on hormone-free MS medium [114]. This phenomenon may possibly be explained by the presence of endogenously produced auxins promoting spontaneous rooting, as suggested by Dore [131]. This theory is supported by the observed effects of age and season on the rooting ability of a species, which in turn is correlated with endogenous auxin level [132]. Similar results have been observed on many other species, including *Cotinus coggygria* [133], *Bambusa vulgaris* [134], and *Cannabis sativa* [135]. 

Auxins are a class of plant hormones that play an important role in cell division, differentiation, and elongation [136]. They promote initiation, growth, and branching of adventitious roots [137]. Rooting of other *Eugenia* in vitro cultures was normally induced by adding auxins to the rooting medium. Pavendan and Rajasekaran [138] used an MS basal medium with 2.5 μM IBA for rooting *E. singampattiana* cultures. Similarly, it was reported that 4.7 μM NAA in 1/5 MS medium was used to root *E. smithii* shoots [116]. Some *Eugenia* species, however, are rooted during acclimatization. For example, *E. uniflora* in vitro cultures were rooted in a pot of autoclaved sand in a nursery with daily watering without applying exogenous auxins [129]. Some *Eugenia* cultures were reported to be acclimatized after in vitro rooting. Gallon, et al. [100] reported acclimatizing rooted *E. uniflora* plantlets in a moistened mixed substrate of vermiculite and sand (1:1), which were kept indoor and under high humidity for a week before gradually decreasing humidity and increasing light. Other acclimatization protocols all follow similar rules, with many *Eugenia* species rooted in a variety of substrates providing they were given sufficient water [100,114,116,138]. 

#### 2.3.4. Summary of Eugenia Tissue Culture and Potential Challenges

A summary of the various in vitro protocols for *Eugenia* species is presented in Table 1. Research on tissue culture of ten *Eugenia* species has shown that all can be successfully cultured in vitro. Viable cultures have been established using seeds, meristems, apical buds, and nodes. Most culture multiplication was based on either full-strength or half-strength MS basal media with the most common hormone combination being BAP and IBA in a common ratio of 1.8:1. Rooting was observed on initiation and multiplication media or hormone-free basal media. NAA was the most used auxin to induce rooting, either used in a dip (9.4 mM) or supplement (0.5 μM–4.7 μM) in reduced-strength MS media. Acclimatization of in vitro-generated plantlets was mostly successful as long as using substrates that drain well and providing shading and regular watering. These techniques may be transferable to other species in the tribe and provide a good starting point for research on threatened wild species. 

Potential challenges and possible solutions demonstrated from these protocols are useful to inform future translations (Figure 1). For threatened species, a main challenge is the limited availability of plant materials, especially those of good quality. In this case, optimizing protocols to directly capture field materials could be beneficial to include genetic diversity and to greatly reduce the required time in protocol, as shown for papaya and grapevine [139,140]. Another potential challenge that may impact multiple stages in tissue culture is endophytes, as shown for *Plinia peruviana* from the Myrteae tribe [101,141]. Meristem culture is a preferred method to reduce endophytes, as well the application of antimicrobials, such as antibiotics, in sterilization process or in media [103,139,142,143]. Moreover, rooting difficulties have been observed on some *Eugenia* species, such as *E. uniflora* [129]. This was improved using temporary immersion systems, which facilitate nutrient uptake and gas exchange, thus benefiting shoot multiplication and rooting simultaneously [144]. Acclimatization is less studied for *Eugenia*, but several papers reported plantlets being sensitive to humidity and waterlogging [83,145]. 

**Table 1 plants-13-02244-t001:** Summary of tissue culture protocols used to propagate *Eugenia* species.

Species	Micro- Propagation Technique	Explant Type and Sterilization Method	Initiation Medium	Multiplication Medium	Rooting Medium	Acclimatization Substrate	References
*E. involucrata*	In vitro germination; Node culture	Seeds; apical buds germinated from seeds (1 cm) 70% ethanol for 1 min, 3% Ca(OCl)_2_ for 15 min, 3% NaOCl for 15 min	Water agar (in vitro germination)	Half-strength MS with 87.6 mM sucrose, 277.5 μM myo-inositol, 0, 0.1 or 0.2 μM BAP; 0, 10, 20 or 30 µM IBA, and 7 g L^−1^ agar, pH 5.8	-	-	[146]
*E. involucrata*	Node culture	Non-woody axillary buds between the third and fourth nodal segments (1 g L^−1^ benomyl and 0.1 g L^−1^ streptomycin sulfate for 30 min) 70% ethanol for 30 s, agitation in 1.5% (*v*/*v*) NaOCl for 15 min, 0.05% (*w*/*v*) HgCl_2_ for 10 min, 1.5% (*v*/*v*) NaOCl solution with three drops of commercial detergent for 10 min; sterile water wash between treatments; explants maintained in 100 mg L^−1^ ascorbic acid	-	Half-strength MS with 87.6 mM sucrose, 277.5 μM myo-inositol, 0.5 µM NAA, 32 µM TDZ, and 7 g L^−1^ agar, pH 5.8	-	-	[82]
*E. involucrata*	Shoot culture	Apical and nodal shoots (1.5 cm) germinated from seeds 70% ethanol, 1.25% NaOCl 15 min, rinse with autoclaved distilled water 3 times, 10 min silver nanoparticles (prepared by reducing 95 mL solution containing 5 mg AgNO_3_ with 5 mL 1% sodium citrate solution, heating in a water bath at 90 °C)	Vermiculite and sand (1:1) substrate, watered daily and fortnightly with a one fourth MS solution (seed germination)	Half-strength MS with half-strength vitamins, 0.5 μM IBA, and 0.9 μM BAP, and 7.0 g L^−1^ agar, pH 5.8	Seedlings developed a root system on multiplication medium	Vermiculite and sand (1:1) substrate in plastic containers, packed into transparent bags to keep moist for indoor acclimatization, then unpacked and placed under shaded environment for outdoor acclimatization	[100]
*E. javanica* (now accepted as *Syzygium aqueum*)	Meristem culture	Meristem (0.5 mm) from a 5-year-old tree 70% ethanol for 15 s, 1% NaOCl for 10 min	Liquid MS with 87.6 mM sucrose and 2.2 μM BAP (4 weeks) followed by MS with 87.6 mM sucrose and 2.2 μM BAP, solidified with agar	MS with 87.6 mM sucrose, 0.4 μM BAP, 0.5 μM NAA, and 8 g L^−1^ agar	9.4 mM NAA dip for 5 s, then placed on half MS with 58.4 mM sucrose and 8 g L^−1^ agar; half- strength MS with 0.5 μM NAA, 58.4 mM sucrose, and 8 g L^−1^ agar	-	[128]
*E. myrtifolia* (now accepted as *Syzygium austral*)	In vitro germination; Shoot culture	Seeds at different maturity level (external integument removed); shoots germinated from seeds 80% ethanol 5 min, 30% (*v*/*v*) commercial bleach 20 min	MS (half- strength macronutrients and full- strength micronutrients) with 2.5 μM TDZ, pH 5.7 (optimal seeds regeneration; in dark) hormone- free half- strength MS, pH 5.7 (adventitious buds elongation)	MS with 4.4 μM BAP, 0.05 μM NAA, and 8 g L^−1^ agar, pH 5.7	Hormone- free MS medium	Sterilized soil and peat (1:1) in clay pots, covered by a glass beaker for 2 weeks, then uncovered and moved to greenhouse for more than 2 weeks before transfer to open air	[114]
*E. myrtifolia* (now accepted as *Syzygium austral*)	Shoot culture	Buds collected in spring 70% ethanol for a few seconds, 50% ACE detergent for 30 min	Half- strength MS with half- strength vitamins, 87.6 mM sucrose, 1 g L^−1^ PVP, 2.2 μM BAP, 0.05 μM IBA, 0.1 μM GA_3_, 56.8 μM filter- sterilized ascorbic acid, and 7 g L^−1^ agar	MS with 87.6 mM sucrose, 1 g L^−1^ PVP, 2.2 μM BAP, 0.05 μM IBA, 0.1 μM GA_3_, 56.8 μM filter-sterilized ascorbic acid, and 7 g L^−1^ agar	-	-	[120]
*E. pyriformis*	Node culture	Nodal segments from adult plants 70% ethanol with Tween 20 (one drop per 100 mL) for 90 s, 1% NaOCl for 20 min	-	MS with 87.6 mM sucrose, 300 mg L^−1^ PVP or ascorbic acid, 2 g L^−1^ activated charcoal, and 5.5 g L^−1^ agar, pH 5.8	-	-	[81]
*E. singampattiana*	Node culture	Nodal segments (0.5–1 cm) 70% ethanol for 1–5 min, 0.1% HgCl_2_ for 3 min	MS with 4.4 μM BAP, 4.5 μM TDZ, 87.6 mM sucrose, and 8 g L^−1^ agar, pH 5.7	-	MS with 2.5 μM IBA, 87.6 mM sucrose, and 8 g L^−1^ agar, pH 5.7	Sterilized garden soil, sand, and vermiculite (2:1:1) in plastic pots covered with polythene bags, watered with a one fourth MS solution every three days, transferred to field conditions after 15 days	[138]
*E. smithii* (now accepted as *Syzygium smithii)*	Shoot tip culture	Apical and nodal shoot tips from a 3-year-old plant; in vitro shoots (0.5 cm) 0.2% HgCl_2_ for 4 min	MS with 58.4 mM sucrose, 2.2 μM BAP, 0.5 μM IBA, and 7 g L^−1^ agar, pH 5.6–5.8	Elongation on MS with 58.4 mM sucrose, 4.4 μM BAP, 2.5 μM IBA, and 7 g L^−1^ agar, pH 5.6–5.8	One-fifth strength MS with 4.7 μM NAA	Garden mould and peat moss (1:1) in plastic pots maintained in a 22–25 °C greenhouse, sprayed with 5 g L^−1^ thiram (fungicide) two to three times	[116]
*E. anthacanthoides* (now accepted as *E. squarrosa)*	Seed culture	Seeds 2.0% NaOCl and Tween 20 for 20 min	Half- strength MS with 3.8 μM thiamine, 555.1 μM myo-inositol, 29.2 μM sucrose, and 2.5 g L^−1^ Gelrite^®^, pH 5.8	-	Seedlings developed roots on initiation medium	Rich substrate in organic matter and abundant watering	[83]
*E. subdisticha*	Seed culture	Seeds 2.0% NaOCl for 20 min	Half- strength MS with 3.8 μM thiamine, 555.1 μM myo-inositol, 29.2 μM sucrose, and 2.5 g L^−1^ Gelrite^®^, pH 5.8	-	Seedlings developed roots on initiation medium	Rich substrate in organic matter and abundant watering	[83]
*E. sulcata*	Seed culture	Seeds 70% ethanol for 1 min, 2.5% NaOCl for 20 min	WPM with activated charcoal, 7 g L^−1^ agar, pH 5.8 ± 0.1	-	Seedlings developed roots on initiation medium	Maintained in 50% shade greenhouse with a micro- sprinkler watering system	[86]
*E. uniflora*	In vitro germination; Node culture	Seeds from wild genotypes; apical and nodal segments (1.5 cm) 70% ethanol for 1 min, 1.25% NaOCl for 25 min	Water agar with 6.0 g L^−1^ agar (in vitro germination); water agar with 87.6 mM sucrose and 7.5 g L^−1^ agar (to verify the existence of endogenous bacteria and fungi)	Half-strength MS with 87.6 mM sucrose, 0.5 μM IBA, 0.9 μM BAP, and 6.0 g L^−1^ agar, pH 5.8	Autoclaved sand in pots and maintained in a nursery environment with daily watering	Sand: organic soil (1:1) and cultivated in a greenhouse with daily watering	[129]
*E. uniflora*	In vitro germination; Shoot tip culture	Seeds from the ripe fruits; apical and axillary shoot tips (0.5 mm) 70% ethanol for 1 min, 0.5% NaOCl for 20 min	WPM with 87.6 mM sucrose and 25 g L^−1^ Gelsan^®^ (in vitro germination)	WPM with 87.6 mM sucrose, 4.44 μM BAP, and 25 g L^−1^ Gelsan^®^ (in conventional system) WPM with 87.6 mM sucrose, 11.1 μM or 17.76 μM BAP, and 25 g L^−1^ Gelsan^®^ (in the natural ventilation system)	-	-	[130]
*E. uniflora*	Shoot tip culture	Shoot segments (2–3 mm) from plants maintained in a greenhouse 70% ethanol for 5 min and 0.3% NaOCl for 10 min	MS with 0.9 μM BAP and 2 g L^−1^ Gelrite^®^ (kept in dark before shoot regeneration)	Half-strength MS (shoot elongation)	Half- strength MS	Sterilized vermiculite in pots covered with a glass beaker, kept in an incubator maintained at 25 °C, 14 h photoperiod, then transferred to a greenhouse after 2 months	[117]
*E. uniflora*	In vitro germination; indirect somatic embryogenesis	Seeds; nodal segments (1.5 cm) 70% ethanol for 10 min, 1.5% NaOCl for 20 min	Water agar with 6.0 g L^−1^ agar (in vitro germination)	Callogenesis on MS with Morel vitamins [147], 555.1 μM myo-inositol, 87.6 mM sucrose, 56.8 μM ascorbic acid, 5.2 μM citric acid, 10.0 μM NAA, 5.0 μM TDZ, and 6 g L^−1^ agar, pH 5.8; somatic embryogenesis on callus induction medium	-	-	[148]
*E. uniflora*	Seed culture	Seeds -	Water agar with 8.0 g L^−1^ agar (seeds germination)	Half-strength liquid WPM added to water agar every 4 weeks	Seedlings developed roots on initiation medium	-	[115]
*E. uniflora*	In vitro germination; seed culture	Ripe seeds (dehydrated for 7 d) 70% ethanol for 30 s, 5.0% NaOCl for 20 min	MS with 66.7 μM GA_3_, 3% sucrose, 8.0 g L^−1^ agar, pH 5.8 ± 0.1	-	-	After 120 d, plants transferred to Basaplant^®^ commercial substrate with Osmocote^®^ 15-09-12 and covered with a polyethylene plastic bag, which was gradually opened after 7 d. The plants were then kept in a nursery with 50% sunlight and watered twice a day.	[149]

MS: Murashige and Skoog medium; WPM: woody plant medium; BAP: 6-benzylaminopurine; IBA: indole-3-butyric acid; NAA: 1-naphthaleneacetic acid; TDZ: thidiazuron; GA_3_: gibberellic acid; PVP: polyvinylpyrrolidone.

## 3. Importance of Tissue Culture in Conservation of Other Myrteae Genera

In situ conservation is a critical method for maintaining genetic resources around the world, while simultaneously protecting the surrounding ecosystems [150]. In Brazilian biomes, where Myrteae is extensively distributed, there have been various in situ conservation units, including national forests and reserves [151]. However, the efficacy of in situ conservation is under pressure from unpredictable extreme weather events and human activities that have led to habitat fragmentation. A 2020 report determined that around 17% of species within the Myrtaceae family in Australia were susceptible to myrtle rust, a proportion expected to increase in the future [152]. Thus, there is an urgent need for complementary conservation methods. In response to this, recent attention has been directed towards the development of ex situ conservation techniques, aimed at preserving the at-risk wild germplasm of the most threatened species [42]. Potentially suitable techniques include seed banking, field-repositories, and tissue culture-based storage including cryopreservation [44]. This portion of the review focuses on Australian Myrteae, given their importance and diversity in Australian ecosystems and the great challenge posed by a pandemic strain of the pathogen causing myrtle rust. A framework for determining the most appropriate conservation action for threatened Australian plants was recently developed and highlighted the need for in vitro technologies to preserve threatened species in the Myrteae [44]. 

### 3.1. Seed Banking

Seed banking is an effective ex situ plant conservation technique utilized across a variety of threatened or valuable plants for long-term germplasm storage [153]. It exploits the natural capacity of seeds to withstand extreme environmental conditions, maintaining embryo viability until conditions are suitable for germination and regeneration [154]. However, not all species are suitable for seed banking, which depends on seed availability and accessibility, capacity of the seed to tolerate dehydration and cold storage, and knowledge of how to germinate the seed after storage [44]. Many Myrteae species display traits associated with poor seed banking potential, such as fleshy fruit and adaptation to rainforest habitats [9,155,156,157]. According to the online Seed Information Database [158], very few Myrteae species have been assessed for their suitability for banking. Of the 31 species that have been assessed, 13 were classified as recalcitrant (intolerant of drying), tentatively suggesting a 42% recalcitrance rate for the tribe, much higher than expected given the general prediction of approximately 95% orthodoxy (tolerance of drying and cold storage) for the Myrtaceae family as a whole [158]. Besides, in a study conducted to assess the exceptionality of Australian native species, seeds of seven species from seven different Myrteae genera (*Archirhodomyrtus*, *Austromyrtus*, *Lenwebbia*, *Pilidiostigma*, *Rhodamnia*, *Rhodomyrtus*, *Uromyrtus*) were reported to be intermediate, which again suggests that seeds from the majority of the tribe may be non-orthodox [44]. For exceptional Myrteae species, conservation methods such as field genebanks and tissue culture-based methods are more suitable.

### 3.2. Field Banking

Field genebanks enable conservation of species or varieties as living plant collections located away from their natural habitats. This method is commonly applied to sexually sterile species (vegetatively propagated), such as banana and cassava, or species that do not produce seeds or produce non-orthodox seeds, such as many tropical fruit trees [159]. Field genebanks allow more convenient access, characterization and assessment of the conserved genetic resources than other ex situ methods [160]. This is especially beneficial for those perennial species that have long life cycles or slow-growing species that otherwise take a long time to be regenerated from seeds [161]. However, long-term conservation is very challenging for living collections in field genebanks. They incur high costs and require trained personnel for their maintenance and are also at risk from environmental stress including extreme weather, fire, flood, plant pests, and diseases. For Myrteae species, field repositories can be helpful in characterizing the genetics and phenotypes of selections useful for further breeding, such as those with agro-industrial potentials or better tolerance of pests and diseases. Only a few Myrteae species with commercial or ecological values have been conserved in field repositories to date, including species from *Feijoa*, *Psidium*, and *Eugenia* [162,163]. Taking guava as an example, field genebanks of various commercial cultivars of common guava have been established in multiple countries, including Brazil, China, and USA, for the purpose of conservation and breeding (Appendix A). While field genebanking has some advantages as mentioned previously, alternative methods are needed for conservation of species that face environmental and resourcing threats to field maintenance. For threatened wild species in the Myrteae, funding to establish and maintain field genebanks is limited and any field repositories established would be at constant risk of infection with *A. psidii*. Tissue culture of these species is a vital complementary tool enabling secure in vitro storage and rapid multiplication of individuals, and facilitating long-term cryopreservation [64]. 

### 3.3. Tissue-Culture-Based Methods for Plant Conservation

In vitro conservation is an essential component of plant conservation methodologies, particularly for exceptional species. Research on *Eugenia* species has demonstrated that tissue culture of other genera in the Myrteae tribe may be feasible and can potentially be accomplished using simple variations of existing techniques. However, as previously mentioned, tissue culture has the limitations of high resource requirements including the need for skilled technicians, specialized infrastructure, and intensive use of consumables and energy [51,61,65]. Tissue culture performed under conditions to rapidly multiply large volumes of plants typically requires subculturing of each explant on a monthly basis. Although highly effective for mass commercial propagation of single species, this is constraining to long-term maintenance of low numbers of many diverse individuals. In order to reduce the resource intensity of in vitro conservation, both slow-growth and cryopreservation methods have been explored, as discussed below. 

#### 3.3.1. Slow-Growth Tissue Culture

Slow-growth tissue culture has been explored as a means of decreasing maintenance costs by reducing the metabolic activity and growth of explants and increasing the length of time between subcultures. A reduction of growth in plant tissue culture is normally achieved by modifying the growing medium, such as by reducing nutrients or adding growth retardants, or by altering growth conditions, such as by decreasing temperature and light intensity [164,165]. Temperature reduction is the most common method used, in some cases paired with a reduction of photoperiod or luminosity, and has been successfully applied to many crops, including banana (*Musa* spp.) [166], pear (*Pyrus* spp.) [167], and potato (*Solanum tuberosum*) [168]. The applicability of this method mainly depends on the degree of cold sensitivity of the plants; therefore, it may not be feasible for tropical species such as pineapple (*Ananas comosus*) [169]. Only limited research has been performed on applying slow growth tissue culture to conserve Myrtaceae species, with *Eucalyptus* one of the most studied genera. For example, *Eucalyptus grandis* can be stored for up to 10 months at room temperature without subculturing using either a full-strength MS hormone-free medium supplemented with 10 mg L^−1^ abscisic acid, or a half-strength MS hormone-free medium [170]. Multiple *Eucalyptus* spp., such as *E. camaldulensis* and *E. blakelyi*, have also been successfully stored at 4–6 °C in the dark for 9 to 29 months [171]. 

Nevertheless, due to the relatively high costs associated with the maintenance of in vitro cultures, slow-growth tissue culture is usually a medium-term conservation strategy. Cryopreservation offers a more suitable approach for long-term conservation as maintenance demands are minimal once the explants are stored in liquid nitrogen. However, as with tissue culture protocol development, effective cryopreservation requires significant research input.

#### 3.3.2. Cryopreservation

Cryopreservation refers to the process of preserving regenerable tissues at a very low temperature, usually in liquid nitrogen (−196 °C). This method is very valuable in conserving genetic resources—for plant species as well as animals—and has also been exploited for storing human tissues for reproductive and clinical purposes [172,173]. There is theoretically no limit to the storage duration of cryopreservation if liquid nitrogen is replenished regularly, because metabolic processes that can negatively affect regeneration capacity are suppressed [174]. Cryopreservation is theoretically applicable to almost all plant species and can use various types of regenerable tissues, including embryos, shoot tips, and pollen [175]. The primary requirement for successful cryopreservation is the inhibition of deadly ice crystal formation during freezing and thawing. This is achieved by reducing cellular moisture content and controlling cooling/warming rates [176,177,178]. 

Explants used for cryopreservation are usually sourced from tissue culture. In this case, successful cryopreservation is thus dependent on the development of suitable tissue culture media and methods [179,180,181]. Selection of explants used in cryopreservation is usually based on the purpose of cryopreservation (e.g., conserving elite genotypes vs. genetic diversity) and is different for different species [182]. Genetic stability is generally guaranteed during clonal cryopreservation for plant species, including transgenic plants, with most cases of instability only associated with the in vitro micropropagation stages [183]. Overall, cryopreservation can bypass the threats facing field genebanks and has lower costs and space requirements than other ex situ conservation methods. However, potential challenges associated with cryopreservation protocol development primarily arise from the fact that cryo-capability is usually species-specific and depends on the health condition of the mother plants [182]. The availability of cryo-capable plant materials for establishing cultures, especially for threatened species, is also usually limited [184]. Moreover, protocol development requires trained personnel and takes time if no closely related species have previously been explored for micropropagation or cryopreservation feasibility [184]. 

The role and central need for cryopreservation in conservation of Myrtaceae species facing the threat of myrtle rust has been recently reviewed in detail [42]. The authors pointed out that cryopreservation provides an ideal alternative to seed banking for ex situ conservation of threatened exceptional Myrtaceae but, with limited successful application within the family, further research may be necessary to fill the knowledge gap and expedite routine adoption of this biotechnology. So far, cryopreservation protocols only exist for approximately 20 species from mainly two genera, including *Eucalyptus* and *Syzygium* [42]. Of these, *Eucalyptus* species are the most explored, with over ten species confirmed for cryo-capability [42,185]. Currently, there are no published cryopreservation protocols available for any Myrteae species. The lack of tissue culture and cryopreservation methodologies for these species is a major research gap. Indeed, our team is currently working to develop methods for the most critically listed species, *Backhousia leptopetala*, *Gossia fragrantissima*, *G. gonoclada*, *G. hillii*, *Lenwebbia* sp. Blackall Range, and *Lenwebbia* sp. Main Range. Preliminary attempts have succeeded in achieving 40% regrowth for apical shoot tips of *G. fragrantissima* with donor cultures cold-pretreated at 10 °C for 2 weeks (unpublished data).

### 3.4. Conservation of Myrteae—Overview

A summary of ex situ conservation status and conservation efforts for threatened Australian Myrteae species is presented in Table 2.

## 4. Conclusions

Myrteae is a significant tribe in terms of socioecological and economic values in the Myrtaceae family. It is currently threatened by myrtle rust in the wild, alongside continued threats of climate and anthropogenic change. Some Myrteae species are some of the worst impacted, with 16 species from the family in Australia alone predicted to be extinct within one generation. With *Eugenia* as a case study, tissue culture has greatly improved propagation of some commercially viable species. The use of vegetative explants makes it possible to clone the plants with desired traits, which is useful in commercial settings. Multiple types of explants, such as seeds of different maturity and quality and nodal segments, have been adopted for tissue culture establishment, which benefits the conservation of threatened species by allowing the extended use of available plant materials. Given not many Myrteae genera have been developed for tissue culture systems, *Eugenia* tissue culture protocols reviewed here can serve as a starting point for propagating and conserving related genera. Ex situ conservation on Australian Myrteae has already been initiated in terms of seed banking, field banking, and tissue culture, while cryopreservation protocols have not been developed yet. A main challenge facing conservation of Myrteae is limited availability of seed for certain species and limited research into seed biology, which is extremely important when informing suitable ex situ methods (Figure 2). As far as available data are concerned, many Myrteae species are ‘non-orthodox’, meaning they are not amenable to seed banking. Given the increasing threats, in vitro methods (tissue culture and cryopreservation) are more suitable for these Myrteae. Specifically, although usually based on the pre-established tissue culture systems, cryopreservation facilitates long-term conservation with less cost, space, and maintenance needed, compared to tissue culture. Significant extra efforts are required to understand seed information for the remaining species as well as validate storage protocols for current ex situ accessions. To complement the current impaired in situ conservation, a clear and urgent need exists for ex situ capture and development of in vitro protocols for the remaining genetic diversity of species that are critically declining in the wild. 

## Figures and Tables

**Figure 1 plants-13-02244-f001:**
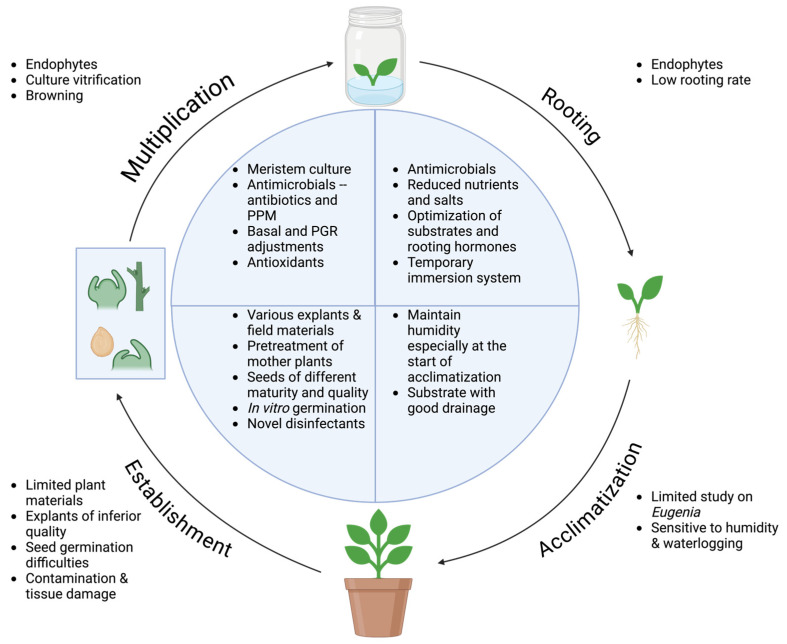
Potential challenges in stages of Myrteae tissue culture (outer circle) and potential solutions (inner circle) proposed based on *Eugenia* tissue culture.

**Figure 2 plants-13-02244-f002:**
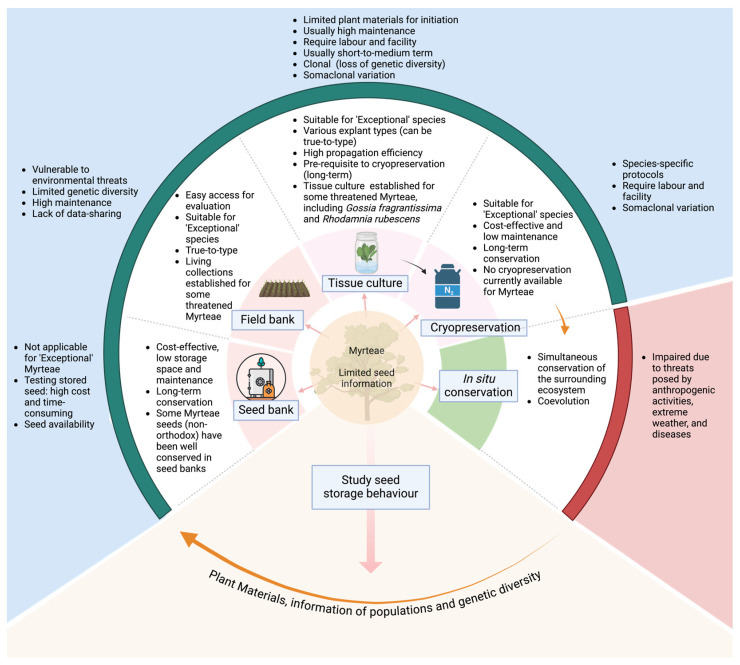
Current conservation efforts for the tribe Myrteae with advantages (inner circle) and disadvantages (outer circle) of each conservation method.

**Table 2 plants-13-02244-t002:** Summary of ex situ conservation efforts for Australian native Myrteae species. Ex situ sites include worldwide seed banking, tissue culture storage, and field banking sites within Australia. They do not include any individual efforts that may be occuring at nurseries around Australia.

Genera	No. Native Species	Threatened Species in AU	Conservation Action in AU	Ex Situ Sites
*Archirhodomyrtus*	1	LC	n/a	Yes
*Austromyrtus*	3	LC	n/a	Yes
*Decaspermum*	2	- *D. struckoilicum*: CR	Conservation advice in effect from 20 March 2023 (conservation advice Decaspermum struckoilicum *Struck Oil myrtle*)	Yes - *D. struckoilicum*: 1
*Eugenia*	1	LC		Yes
*Gossia*	20	- *G. fragrantissima*: EN - *G. gonoclada*: EN - *G. inophloia*: CR (QLD) - *G. acmenoides*: EN population (NSW)	*G. fragrantissima*: conservation advice in effect from 16 July 2000, Border Ranges Rainforest Biodiversity Management Plan—NSW and Queensland 2010; *G. gonoclada*: conservation advice *Gossia gonoclada* angle-stemmed myrtle 2016, *Gossia gonoclada* recovery plan 2019–2029	Yes - *G. fragrantissima*: 6 (four seed accessions in Australian PlantBank) - *G. gonoclada*: 1 - *G. inophloia*: 10 - *G. acmenoides*: 8
*Lenwebbia*	2	- *Lenwebbia* sp. *Main Range*: CR - *Lenwebbia* sp. *Blackall Range*: EN (QLD)	*Lenwebbia* sp. *Main Range*: conservation advice in effect from 22 April 2022	Yes - *Lenwebbia sp. Main Range*: 1
*Lithomyrtus*	11 (All native to AU)	- *L. linariifolia*: V (NT)		Yes *L. linariifolia*:1
*Pilidiostigma*	6	LC		Yes
*Rhodamnia*	20	- *R. angustifolia*: CR - *R. longisepala*: CR *- R. maideniana*: CR - *R. rubescens*: CR - *R. arenaria*: EN (QLD)	Conservation advice in effect (*R. angustifolia*; *R. longisepala*; *R. maideniana; R. rubescens*)	Yes - *R. angustifolia*: 0 - *R. longisepala*: 1 *- R. maideniana*: 8 (one seed accession in PlantBank) - *R. rubescens*: 10 (three seed accessions in PlantBank and Brisbane Botanic Gardens Conservation Seedbank)
*Rhodomyrtus*	6	- *R. psidioides*: CR	Conservation advice in effect from 12 December 2020	Yes - *R. psidioides*: 16 (six seed accessions in PlantBank, Australian National Botanic Gardens Seed Bank, and Brisbane Botanic Gardens Conservation Seedbank)
*Uromyrtus*	4	- *U. australis*: EN	Conservation advice in effect from 23 November 2021 and Border Ranges Rainforest Biodiversity Management Plan—NSW and Queensland 2010	Yes - *U. australis*: 9 (one seed accession in PlantBank)
Total	76			

Data retrieved from World Checklist of Selected Plant families (WCSP), Botanic Gardens Conservation International (BGCI) ThreatSearch, and BGCI PlantSearch. Conservation status sourced according to Biodiversity Conservation Act 2016, Nature Conservation (Plants) Regulation 2020, Territory Parks and Wildlife Conservation Act 1976, and the Environment Protection and Biodiversity Conservation Act 1999 (EPBC Act). Seed bank data were retrieved from Australian Seed Bank Partnership. LC: Least Concern; NT: Near Threatened; VU: Vulnerable; EN: Endangered; CR: Critically Endangered.

## Data Availability

Not applicable.

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
