# Peer review of "Tissue Culture Innovations for Propagation and Conservation of Myrteae—A Globally Important Myrtaceae Tribe"

_plants, 2024, doi:10.3390/plants13162244_

Round 1
Reviewer 1 Report
Comments and Suggestions for Authors
Myrteae species benefit the human society for their wide distribution and socioeconomic and ecological values. Some of them have been increasingly threatened by anthropogenic activities, fungal pathogen or endangered therefore tissue culture is of great importance.This review gave a through summary tissue culture technique of the reported plants from Myrteae. The technology of tissue culture could help them for micropropagation and conservation. Systematic progress about the tissue culture of Myrteae, especially Eugenia species, were reviewed in this study. The paper is integrated and valuable. However, the paper specifically focused of the tissue culture of Myrteae. The 1 introduction and the 2.1 Introduction to micropropagation are straightforward or some basic knowledge could be deleted. The same on the full text, many sentence are repeated. The abstract might be not so clear to clarify for the main contents.
I recommend the paper a minor revision before the final publication.
Comments on the Quality of English LanguageThe language is well writen .
Reviewer 2 Report
Comments and Suggestions for Authors
This manuscript presents an extremely thorough and cohesive review of the state of tissue culture and in vitro conservation of the Myrteae. The authors present a comprehensive review of the conservation and threat status of Myrteae species and lay out an excellent summary of the in vitro techniques currently available to propagate and preserve the species. The review is well-written and provides a wealth of information on a topic that is deeply important to the plant conservation community. The review is clearly thoroughly researched and will be of use to many conservation practitioners working in this globally important species.
Specific comments:
Line 148: “Micropropagation” is misspelled in the title for section 2.2.
Line 182: “Species” is not italicized in the title for section 2.3.
Line 207: “include seeds, embryos, shoots, axillary or nodal buds, and meristem”: Please change “meristem” to “meristems” or “meristematic tissues”, or another plural term.
Line 221: “such as shoots, meristem, and nodes”: Please use a plural form of meristem.
Lines 224-226: Are there any examples that could be cited of species in the Myrteae where shoots produced from callus differ genetically from the mother plant?
Line 317: “as suggested by [129]”: Please include names of authors before citation number.
Table 1: Can the headings of this table be spaced in such a way as to avoid splitting words into two separate lines (e.g. “me-dium” split into two lines in columns 5 & 6 could be spaced to keep “medium” on one line). In addition, it may be helpful to the reader to repeat the column titles at the beginning of each additional page of Table 1, to avoid scrolling back to line 348 to see the titles.
Line 497: “species is presented in Error! Reference source not found..”: Please fix the reference.
Table 2: Please reference this table somewhere in the text.
Line 514: Please change “needs” to “need”.
Lines 502-516: The Conclusions section is reading like a laundry list of statements, rather than a cohesive wrap-up of the review presented. Given the excellent writing and thorough presentation of the literature throughout the rest of the manuscript, the Conclusions could be tightened up to more cohesively tie together the threats to Myrteae species and the progress in their propagation and conservation.
Line 525: “Not. Applicable.” should be changed to “Not applicable”.
Reviewer 3 Report
Comments and Suggestions for Authors
The authors have collected a nice amount of information on the use of tissue culture for the propagation and conservation of Myrteae in this nice and interesting review. This gathered information will help researchers working on this and related species. I suggest adding new important topics within each section that would further enrich this document. As detailed below, some sentences need references to support the information. I miss an illustration showing what this species looks like and even an illustration showing the different stages of micropropagation procedures. Also, a flowchart on the conservation efforts of Myrteae/Myrtaceae would catch the attention of the audience and enrich the review paper.
I have included below a detailed review report with suggestions, questions, and points for clarification:
L 24-25: … largest angiosperm family, is represented by about 6000
L 26: to warm temperate regions, centered in
L 28: … about 2500 species
L 29: are very diverse, but
L 35: fruits of varying sizes throughout the year for frugivorous
L 41: …Brazilian cherry (add a reference to support this sentence)
L 41-43: Myrtus communis, a cultivated ornamental plant with multiple economic values, is a species valued by the food and pharmaceutical industries for its culinary and medicinal properties [14].
L 45-46: traditional medicine (add a reference to support this sentence)
L 54: as well as natural disturbance (add references to support this sentence)
L 63: Myrtle rust (add a reference to support this sentence)
L 79: from the genera Myrcia and Eugenia (add a reference to support this sentence)
L 81: conservation
L 82: is proving to be essential to preserve these species (add references to support this sentence – there are some nice recent paper on this topic – see the following suggestion: https://doi.org/10.3390/plants11081017
L 85-86: Suggestion to enjoy this sentence to mention about the propagule choice for tissue culture or cryo - see my suggestion next and also a reference to support this sentence.
Long-term ex situ conservation therefore requires other methods, such as tissue culture micropropagation and cryopreservation, and the propagule choice depends on whether the genes or the specific genetic combination in an elite tree are the program conservation targets https://doi.org/10.1007/s11240-020-01846-x .
L 94 and L 103: The same word is spelled two different ways - "utilised" and "utilized" - please check it along the manuscript.
L 136-137: are not widely available for all woody plants due
L 143: Reference needed
L 170: add a reference on a paper using cutting propagation – see a suggestion for a recent paper on cutting propagation on a critically endangered Myrtaceae species - https://doi.org/10.1080/0028825X.2022.2158110
L 183: Consider adding as first topic 2.3.1 Selection of propagules for in vitro initiation. So you can consider writing when it is beneficial to use seeds or clonal propagation.
L 188-191: How about field-initiated propagules such as seeds or node segments? Double check the sentence on lines 188-191, I think these are the ideal conditions and not the most used - also please add a reference to support this.
L 191: They are often pretreated
L 193: before initiating nodal cultures harvested from xxxx.
L 195: to establish xxxx (nodal sections? Seeds?) of E. involucrata cultures
L 197: Add more details, such as concentration and when the application was performed.
L 201: reference needed
L 204: bleach solution (Add a range of the most commonly used concentrations) or other chemical disinfectants such as xxx, xxxx to eliminate
L 209-210: please explore it further - you can refer to collect large part of the population instead of clones - add a reference then.
L 218: Another example for your consideration- In the research paper on swamp maize propagation that I suggested as a reference, the authors found that de-pulping the seeds was critical to shortening the mean time to germination and positively affected the germination percentage.
L 222-224: reference needed
L 231: I miss here one of the main challenges when using field growth segments for clonal initiation - I mean contamination x oxidation - consider covering this point.
L 237: ….10 to 25 minutes immersion - references needed
L 238: small amounts (please add the amount)
L 242-244: It is not clear - so they germinated the seeds in vitro and then collected the segment and only after that they did the sterilization - is that right?
L 258: reported the use of a MS basal medium
L 259: micronutrients,
L 270: For example, Toussaint et al [114] reported the use of MS
L 271: indole-3-butyric acid (IBA)
L 284: How about the use of substances (for example, plant preservative mixture, etc.) to help alleviate endophyte contamination? I strongly suggest a sentence about the major challenges posed by endophytes
L 291: So add a sentence about when these propagules (meristems) are used - if the main goal is to produce virus-free plants, is there any work on Myrtaceas? Please explore this further and add references.
L 294-296: Reference needed
L 297-298: The most commonly used pair of plant growth regulators is BAP and IBA, commonly combined in a ratio of 1.8:1 [101,114,127].
L 299: Long et al. [118]
L 314: delete the “dot” after media
L 317: as suggested by “add the author surname and if et al. if needed” [129].
L 322-323: reference required
L 324: was normally
L 328: For example, E. uniflora in vitro cultures were rooted
L 330: “Other acclimatization protocols all follow similar rules” - you were talking about rooting in the previous sentence, so it is necessary to further explore the acclimatization phase
L 334: I strongly suggest the addition of a new topic before the summary describing the main challenges in micropropagation of this species - this is a really important information for the review and author can cover since endophytes to difficult to root species. Also, I strongly suggest the inclusion of a flowchart describing the main steps in the micropropagation of this species, and the author could add the main challenges in each phase.
L 334: “Summary of ?”
L 345: I strongly suggest a sentence mentioning how long it takes each phase of a basic micropropagation protocol for this species - author can consider using the most popular explants for initiation.]
Table 1 – describe all the abbreviation used in the table in the table footer
L 351-353: Reviewing sentence grammar: In situ conservation is the predominant method of conserving genetic resources worldwide because of the benefits of synergistic protection of associated ecosystems
L 351: In situ?? - check this information.
L 362-363: References needed
L 384: …. as a whole (reference needed)
L 390: Add the reason
L 401: double check the page numbers - they need formatting
L 406-408: references needed
L 408-410: Where is it being performed? Country?
L 411: some advantages as mentioned previously,
L 412: tools or methods?
L 4515-417: references needed
L 420: toolkit??
L 423: Reference needed
L 445: For example,
L 445: was stored for up
L 445-447: Is this not an example of slow growth? I could not see what the slow growth conditions were on the sentence - sentence needs checking.
L 448-449: Explore this information further - what were the growing conditions and how long could this species be maintained without subculturing?
L 470: including embryos, shoot tips, and pollen (References needed)
L 474: during clonal cryopreservation
L 482: further collaboration??
L 484: “20 species from mainly two genera, including Eucalyptus and Syzygium” – So add some references or even a table showing this - it would be another good piece of information to help readers.
L 493: I strongly suggest adding a paragraph on the challenges of implementing cryopreservation procedures in general and specific to this species.
L 497: “in Error! Reference source not found..” Please review this…
Table 2 - In the third table column is written Australia and in the fourth column only Au has been added - please use the same way in all cases.
L 500: Again, I think a figure/flowchart on the conservation efforts of Myrteae/Myrtaceae would catch the attention of the audience and enrich the review paper.
Also, I really miss an illustration showing what this species looks like and even an illustration showing the different stages of micropropagation procedures.
Before the conclusion, a topic on future efforts of propagation/conservation of this species would further enrich this review paper.
Round 2
Reviewer 3 Report
Comments and Suggestions for Authors
A great review job has been done covering all the points/suggestions mentioned. I strongly support the publication of this nice and informative review.